# Catalytic mechanism of the zinc-dependent MutL endonuclease reaction

Kenji Fukui[1] , Tatsuya Yamamoto[2] , Takeshi Murakawa[1], Seiki Baba[3], Takashi Kumasaka[3] , Takato Yano[1]

**DNA mismatch repair endonuclease MutL binds two zinc ions. However, the endonuclease activity of MutL is drastically enhanced by other divalent metals such as manganese, implying that MutL binds another catalytic metal at some site other than the zinc-binding sites. Here, we solved the crystal structure of the endonuclease domain of *Aquifex aeolicus* MutL in the manganese- or cadmium-bound form, revealing that these metals compete with zinc at the same sites. Mass spectrometry revealed that the MutL yielded 5′-phosphate and 3′-OH products, which is characteristic of the two-metal-ion mechanism. Crystallographic analyses also showed that the position and flexibility of a highly conserved Arg of *A. aeolicus* MutL altered depending on the presence of zinc/manganese or the specific inhibitor cadmium. Site-directed mutagenesis revealed that the Arg was critical for the catalysis. We propose that zinc ion and its binding sites are physiologically of catalytic importance and that the two-metal-ion mechanism works in the reaction, where the Arg plays a catalytic role. Our results also provide a mechanistic insight into the inhibitory effect of a mutagen/carcinogen, cadmium, on MutL.**

## Introduction

DNA mismatch repair (MMR) system corrects DNA mismatches generated by errors in DNA replication, recombination among non-identical sequences, and other processes (1, 2, 3, 4, 5). Because MMR significantly contributes to maintain genome integrity, mutations in bacterial MMR genes are related to an increase in antibiotic resistance (6), and mutations and epigenetic silencing of human MMR genes can be causative of a hereditary cancer, Lynch syndrome (7, 8, 9).

In the MMR pathway, bacterial MutS and its eukaryotic counterparts (MutSα and MutSβ) recognize DNA mismatches and subsequently recruit bacterial MutL and eukaryotic counterparts such as MutLα, respectively (3, 10, 11, 12, 13). MutLs, with a few exceptions, are endonucleases that incise the error-containing (i.e., newly synthesized) strand of the mismatched duplex (14, 15, 16, 17). Prokaryotic MutL endonucleases are homodimers, whereas eukaryotic MutL endonucleases are heterodimers, with only one of the two subunits having endonuclease activity. The bacterial DNA polymerase III β subunit (β-clamp) and its eukaryotic counterpart PCNA interact with MutL, which directs the incision to the newly synthesized strand (18, 19, 20, 21). MutL-dependent incision induces downstream MMR reactions, where single-stranded DNA-specific exonucleases excise the error-containing strand to generate a gap region, replicative DNA polymerases fill the gap, and DNA ligases seal the nick to complete the reaction.

Prokaryotic MutL endonucleases and the endonuclease subunits of eukaryotic MutL homologs include two domains: the N-terminal ATPase domain and the C-terminal endonuclease domain (CTD) (22, 23). Crystallographic analyses have revealed that CTD consists of the regulatory and dimerization subdomains. The regulatory subdomain is missing in MutLs from some bacterial species including *Aquifex aeolicus*, whereas the dimerization subdomain is found in all MutL endonucleases (24). The dimerization subdomain contains the metal-binding sites, in which multiple zinc ions are coordinated by the residues from highly conserved motifs: DQHA$X_2$E$X_4$E, ACR, and CPHGRP (underlined residues coordinate the zinc ions) (23, 25, 26, 27, 28). These motifs are conserved in all MutL endonucleases including eukaryotic MutLγ, another MutL homolog that participates in both MMR and crossover formation during meiosis (29, 30, 31, 32), but not conserved in MutLs without endonuclease activity (33, 34).

Human and yeast MutLα endonucleases exhibit manganese-dependent activation (14, 15). Similar finding was observed with MutLs from *Bacillus subtilis*, *Thermus thermophilus*, *A. aeolicus*, and other bacteria (23, 35, 36). In the crystal structure of the *B. subtilis* MutL CTD, zinc ions were observed only when the crystallization condition contained zinc ions (23). One of the two binding sites had a lower affinity to zinc ion and was presumed to have a structural function rather than a catalytic one. This was later supported by structural analysis of the *A. aeolicus* MutL (aqMutL) CTD in its single or multiple zinc-bound states, where multiple zinc ions maintained the conformation of the CTD (37). It has been confirmed that the Asp residue at the beginning of the DQHA$X_2$E$X_4$E motif is critical for the nuclease activity of eukaryotic and bacterial MutLs (14, 23, 35),

[1]Department of Biochemistry, Faculty of Medicine, Osaka Medical and Pharmaceutical University, Takatsuki, Japan   [2]Bioorganic Research Institute, Suntory Foundation for Life Sciences, Kyoto, Japan   [3]Structural Biology Division, Japan Synchrotron Radiation Research Institute (JASRI), Hyogo, Japan

Correspondence: kenji.fukui@ompu.ac.jp; takato.yano@ompu.ac.jp

although the Asp residue is not involved in zinc binding. Therefore, it is inferred that another metal ion, such as a manganese ion, acts as a catalytic metal ion and is coordinated by the Asp residue (38).

On the other hand, Gueneau et al have proposed another model based on the crystal structure analyses of the yeast MutL$\alpha$ CTD (38). The yeast MutL$\alpha$ CTD bound zinc ions at two binding sites even without adding zinc ions during the protein preparation and crystallization procedures (25). The arrangement of the two zinc ions is reminiscent of the two-metal-ion mechanism that is widely adopted in phosphoryl transfer enzymes. The same arrangement was also observed for the yeast MutL$\gamma$ CTD (39). These imply that the two zinc ions are the catalytic metal ions for MutL endonucleases. This hypothesis is supported by the recent finding that human MutL$\alpha$ exhibited the nuclease activity with zinc ions alone (40). However, the result does not rule out the possibility that a catalytic zinc ion binds to an unknown site such as the Asp residue of the DQHAX$_2$EX$_4$E motif.

Cadmium exhibits chemical similarities to zinc and is known to replace zinc in many metalloenzymes (41). Incorporation and accumulation of cadmium within human bodies is associated with several types of cancer; therefore, cadmium is classified as a carcinogen for human (42). In addition, the mutagenic effect of cadmium on mammalian cells has also been reported (43). The mutagenic effect of cadmium has been linked to specific inhibition of the MMR activity (44). Recently, we found that cadmium ions bind to the same sites of the aqMutL CTD as zinc ions (27, 37). Furthermore, Sherrer et al revealed that cadmium ion specifically interfered with the endonuclease activity of human MutL$\alpha$ (40). However, the detailed mechanism of inhibition by cadmium remains unclear as the catalytic mechanism of the nuclease activity of MutL is not yet understood.

In this study, we first solved the crystal structure of the aqMutL CTD in the manganese ion-bound state. Manganese ions were located at the same binding sites as zinc ions, which indicates that the nuclease catalytic site is defined by these zinc-binding sites. Mass spectrometric analysis revealed that aqMutL yielded the 5'-phosphate and 3'-OH products, which is characteristic of the two-metal-ion mechanism. By comparing the zinc- and cadmium-bound structures of the aqMutL CTD, we found that the position and flexibility of the Arg residue of the CPHGRP motif were affected by binding cadmium ions. Site-directed mutagenesis and an in vivo complementation assay indicated that the Arg residue plays a catalytic role in the nuclease reaction. From these and other results, we propose that MutL incises DNA by the two-metal-ion mechanism, where the positive charge of the Arg residue of the CPHGRP motif is catalytically important. The mutagenicity of cadmium might be explained by the cadmium-dependent miscoordination of the catalytic Arg residue.

## Results and Discussion

### Manganese ions bind to the same sites of aqMutL as zinc ions

Previously, we determined the crystal structure of the zinc ion-bound form of the aqMutL CTD, where three zinc ions were identified at the M1, M2, and M3 sites (Fig 1A and B) (37). The M1 and M2 sites are defined by the DQHAX$_2$EX$_4$E, ACR, and CPHGRP motifs and are identical to the zinc ion-binding sites of the yeast MutL$\alpha$ and *B. subtilis* MutL CTDs (23, 25). The M3 site is unique to aqMutL and required to maintain the open conformation of the active site (37).

The observations of the manganese ion-dependent activation, the findings supporting structural functions for zinc ion, and the presence of a critical Asp residue outside the M1–M3 sites imply the possibility that MutL has another site accommodating a catalytic metal ion. In order to explore this possibility, we performed crystal structure analysis of the aqMutL CTD complexed with manganese ions. The aqMutL CTD, which is purified as the zinc-bound form, was incubated with the buffer solution containing 2 mM manganese ion at 70°C for 24 h. After dialysis against the same buffer at 25°C for 24 h, the absence of zinc was confirmed by inductively coupled plasma-atomic emission spectrometry (ICP-AES). The protein was concentrated and crystallized in the presence of a 50 mM manganese ion. The presence of manganese ions and the absence of zinc ions in the crystal were confirmed by the manganese K-edge and the zinc K-edge X-ray absorption fine structure analyses, respectively (Fig S1A and B). The crystal structure was solved with manganese single-wavelength anomalous dispersion phasing (Table S1). Peaks of electron density in the anomalous diffraction map calculated from data collected at a wavelength of 1.8900 Å were clearly observed at the M1, M2, and M3 sites of the aqMutL CTD at 100% occupancy (Fig 1C). The wavelength of 1.8900 Å was chosen to be above the absorption edge of manganese so that the observed peaks of anomalous diffraction correspond to manganese atoms. The overall structure of the manganese-bound form was almost identical to that of the zinc-bound form; the root-mean-square deviation between the two structures was 0.1 Å (Fig 1D).

No peak of electron density in the anomalous diffraction map was observed around Asp351 of the DQHAX$_2$EX$_4$E motif. Additional peaks were found at the other sites M4, M5, and M6 that were coordinated by Asp366, Glu384, and Glu416, respectively (Fig S2A). These acidic residues are not conserved in other MutL homologs and, in addition, were confirmed to not be required for the manganese-dependent endonuclease activity of aqMutL (Fig S2B). Therefore, metal ions at the M4, M5, and M6 sites are not the catalytic metal ions, which would be artifacts observed because of high concentrations of manganese ions in the preincubation and crystallization conditions. Thus, it was confirmed that manganese ions bind to the same sites of aqMutL as zinc ions. Together with experimental results from previous studies identifying catalytically important residues (14, 15, 34), the M1 and M2 sites are likely to be the binding sites for the catalytic metals.

As Gueneau et al have pointed out, arrangements of the M1 and M2 zinc ions of MutL are similar to those of nucleases that employ the two-metal-ion mechanism (25, 38). To examine the possibility that the two-metal-ion mechanism works in the MutL endonuclease reaction, properties of the products of the aqMutL nuclease reaction were characterized. The 16-bp double-stranded DNA was degraded by aqMutL (Fig 2A), and the reaction products were purified and analyzed by MALDI TOF/TOF mass spectrometry (MS). As shown in Fig 2B, a series of degradation fragments was detected. The cleavage products were completely dissociated into single-

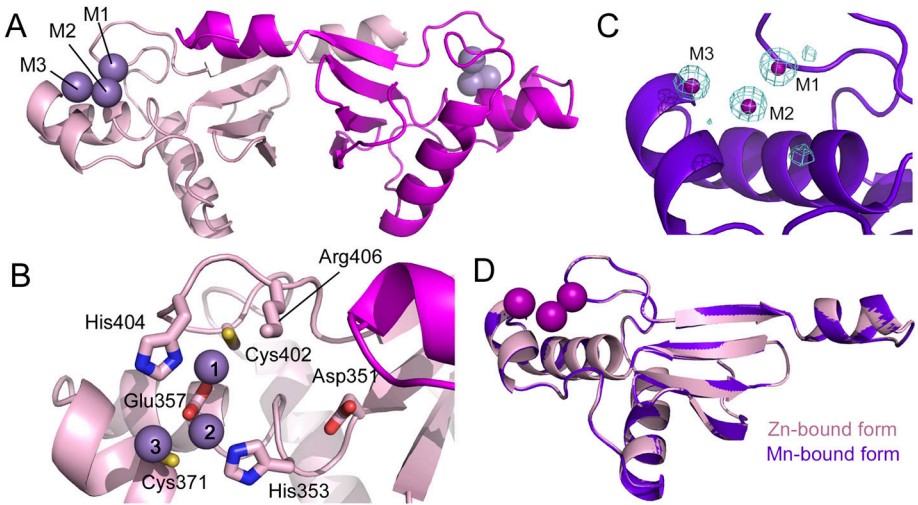

**Figure 1. Manganese ions bind to the same sites as zinc ions.**
**(A)** Crystal structure of the zinc-bound form of the *Aquifex aeolicus* MutL (aqMutL) C-terminal endonuclease domain (CTD). The two subunits of the dimer are shown in light pink and magenta. Zinc ions at M1, M2, and M3 sites are shown in sphere models. **(B)** The zinc-binding site of the aqMutL CTD. The M1, M2, and M3 zinc ions are coordinated by the residues from the conserved motifs. The Asp residue (Asp351) of the $DQHAX_2EX_4E$ motif is not involved in the zinc coordination. **(C)** In the manganese-bound form of the aqMutL CTD, three manganese ions were found at the same sites as zinc ions. Mesh represents anomalous difference density calculated from data collected at 1.89 Å. The density is shown at 3.0 σ. **(D)** Superimposition of the overall structures of the zinc-bound (light pink) and manganese-bound (purple) forms of the aqMutL CTD. The Cα root-mean-square deviation between the two structures was 0.1 Å. Metal ions are shown in sphere models. Single subunit of the dimer is shown.

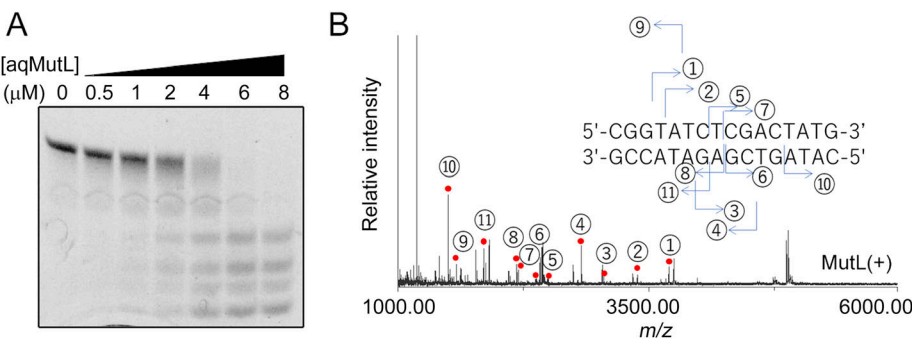

**Figure 2. Mass spectrometric analysis of the *A. aeolicus* MutL (aqMutL) endonucleolytic products.**
**(A)** Electrophoretic analysis of the DNA fragments generated by aqMutL. The 16-bp double-stranded DNA was incubated with 0–8 μM aqMutL. The generated DNA fragments were separated by 25% polyacrylamide gel electrophoresis and stained with SYBR Gold. **(B)** Mass spectrum of the products generated by the aqMutL endonucleolytic reaction. The 16-bp double-stranded DNA was incubated with (MutL(+)) or without 1.9 μM aqMutL. The reaction products were purified and analyzed by MALDI TOF/TOF mass spectrometry. Only the spectrum for the MutL(+) sample is shown. Red dots indicate peaks observed only for the MutL(+) sample. The inset indicates the nucleotide sequence of the substrate 16-bp double-stranded DNA that was degraded by aqMutL. Numbering above the peaks with red dots indicates each fragment represented in the inset. For example, the peak 1 corresponds to the fragment 5'-TATCTCGACTATG-3'.

**Table 1. Theoretical and observed masses of the DNA fragments generated by *A. aeolicus* MutL digestion.**

| Fragment no. | Group at the cleaved site[a] | Theoretical mass | Observed mass | ΔMass[b] | Error (ppm)[c] |
|---|---|---|---|---|---|
| 1 | 5'-phosphate | 3,699.62 | 3,699.93 | 0.31 | 83.92 |
| 2 | 5'-phosphate | 3,386.56 | 3,386.83 | 0.27 | 79.16 |
| 3 | 3'-hydroxy | 3,051.57 | 3,051.82 | 0.25 | 82.82 |
| 4 | 5'-phosphate | 2,827.49 | 2,827.73 | 0.24 | 85.75 |
| 5 | 5'-phosphate | 2,489.42 | 2,489.66 | 0.24 | 95.02 |
| 6 | 3'-hydroxy | 2,409.46 | 2,409.68 | 0.22 | 92.50 |
| 7 | 5'-phosphate | 2,200.38 | 2,200.58 | 0.20 | 92.22 |
| 8 | 5'-phosphate | 2,185.38 | 2,185.57 | 0.19 | 88.12 |
| 9 | 3'-hydroxy | 1,503.31 | 1,503.47 | 0.14 | 95.05 |
| 10 | 3'-hydroxy | 1,158.26 | 1,158.33 | 0.07 | 60.71 |
| 11 | 5'-phosphate | 1,856.33 | 1,856.50 | 0.18 | 94.32 |

[a]The groups at the terminus of the fragment, which were created by the MutL endonucleolytic reaction.
[b]Difference between theoretical and observed masses.
[c]Each fragment was identified with an error of less than 100 ppm.

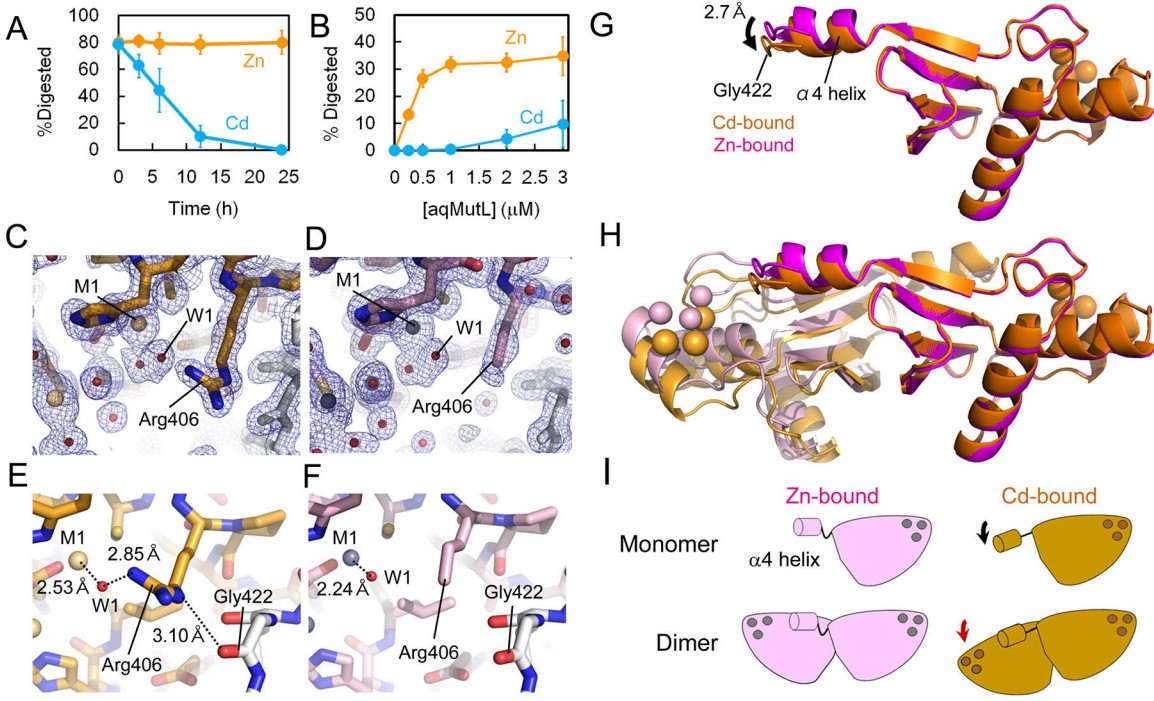

**Figure 3. Cadmium ion is an inhibitor of the *A. aeolicus* MutL (aqMutL) endonuclease activity.**
**(A)** aqMutL was incubated in the 0.5 mM CdCl₂- or ZnCl₂-containing buffer at 60°C for 0, 3, 6, 12, or 24 h. The endonuclease activity of the CdCl₂- or ZnCl₂-treated aqMutL (2 μM) was measured in the divalent metal-free buffer using a supercoiled plasmid DNA as the substrate at 70°C for 30 min. Average values from three independent experiments are shown with standard deviations. **(B)** After treatment with 0.5 mM ZnCl₂ or CdCl₂ at 60°C for 12 h, the endonuclease activity of aqMutL was measured at different protein concentrations in the divalent metal-free buffer at 70°C for 15 min. Average values from three independent experiments are shown with standard deviations. **(C)** The electron density of the side chain of Arg406 is obvious in the cadmium-bound form of the aqMutL C-terminal endonuclease domain (CTD). Blue mesh represents the $2F_o–F_c$ map shown at 3.0 σ. Gold and red spheres represent cadmium ions and water molecules, respectively. **(D)** The electron density of the guanidino group of Arg406 is missing in the zinc-bound form of the aqMutL CTD. Blue mesh represents the $2F_o–F_c$ map shown at 3.0 σ. **(E)** The guanidino group of Arg406 is fixed by a cadmium ion via the W1 water molecule in the cadmium-bound form. And Nε of Arg406 makes a hydrogen bond with the main-chain carbonyl group of Gly422. **(F)** No interactions involving the side chain of Arg406 were observed in the zinc-bound form. **(G)** Superimposition of the monomeric structures of the cadmium-bound (orange) and zinc-bound (magenta) forms of the aqMutL CTD. The cadmium-bound and zinc-bound forms were superimposed so that the positions of the three metals and the metal-coordinating residues of the subunit on the right side coincide between the two forms. Metal ions are shown in sphere models. Gly422 is located at the tip of the loop structure extending from the α4 helix of the other subunit. The α4 helix in the cadmium-bound form moved by 2.70 Å compared with that in the zinc-bound form. **(H)** Superimposition of the dimeric structures of the cadmium-bound (gold/orange) and zinc-bound (light pink/magenta) forms of the aqMutL CTD. **(C, E, F)** It should be noted that, of the two catalytic sites shown here, the one on the right side (indicated by a red arrow head) is shown in panels (C, E, F). **(I)** A schematic representation of cadmium-dependent structural changes in the aqMutL CTD, which depicts the movement of the α4 helix (black arrow) and the change in the subunit arrangement (red arrow).
Source data are available for this figure.

stranded DNAs during the MS analysis. The masses of the detected fragments were consistent with the theoretical masses with errors of less than 100 ppm (Table 1), assuming that aqMutL leaves 5′-phosphate and 3′-OH groups at the cleavage site. Although fragments with a 5′-phosphate group could not be efficiently fragmented by our MS/MS analysis, those with a 3′-OH group were precisely identified by the MS/MS analysis (Fig S3A–C). Thus, it was revealed that aqMutL digested the DNA at the 3′-side of the phosphodiester bond. This is characteristic of cleavage by the two-metal-ion mechanism (45).

## Cadmium ion limits the flexibility of the side chain of Arg406

In the reaction by the two-metal-ion mechanism, two metal ions act as a general base and a Lewis acid to activate the attacking water molecule and stabilize the negatively charged transition state, respectively (46, 47). In addition to amino acid residues involved in

the coordination of these catalytic metal ions, other amino acid residues are often involved in the catalysis by assisting the functions of the metal ions. In order to explore the possible catalytic residues of MutL, we focused on the inhibitory effect of cadmium on MutL endonuclease activity. It was recently found that cadmium was a specific inhibitor of the endonuclease activity of human MutLα by competing with zinc ions (40), which would be one of the mechanisms of the carcinogenic effect of cadmium.

To test whether cadmium interferes with the endonuclease activity of aqMutL, zinc ions in the purified aqMutL sample were replaced with cadmium ions. The protein solution was incubated in the buffer containing 0.5 mM zinc or cadmium ions at 60°C for various periods of time, and the endonuclease activity of aqMutL was measured. As shown in Fig 3A and B, the activity was significantly reduced after 12 h incubation with cadmium ion, indicating that cadmium ions gradually replaced zinc ions and interfered with the endonuclease activity of aqMutL. The DNA binding of aqMutL

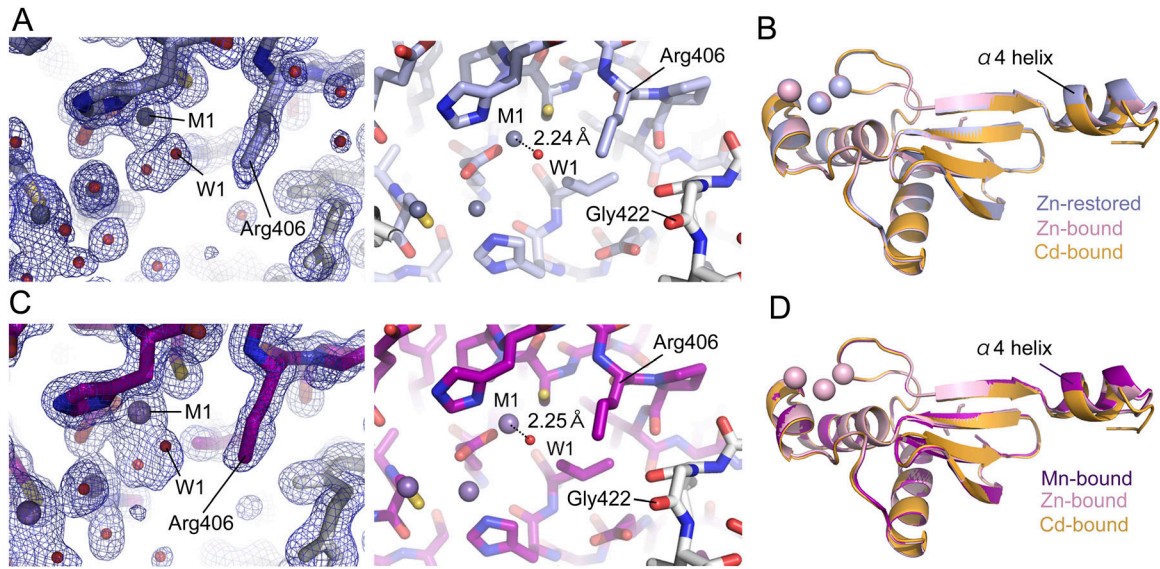

**Figure 4. Metal ion-dependent conformational changes of the *A. aeolicus* MutL (aqMutL) C-terminal endonuclease domain (CTD).**
**(A)** The catalytic site of the zinc-restored aqMutL CTD, in which cadmium ions in the crystal were replaced with zinc ions. Blue mesh represents the $2F_o–F_c$ map shown at 3.0 $\sigma$. Gray and red spheres represent zinc ions and water molecules, respectively. The guanidino group of Arg406 could not be identified. **(B)** Superimposition of the overall structures of the zinc-restored, zinc-bound (cocrystallized with zinc ions), and cadmium-bound (cocrystallized with cadmium ions) forms of the aqMutL CTD. **(C)** The catalytic site of the manganese-bound form of the aqMutL CTD. Blue mesh represents the $2F_o–F_c$ map shown at 3.0 $\sigma$. Purple spheres indicate manganese ions. The guanidino group of Arg406 could not be identified. **(D)** Superimposition of the overall structures of the manganese-bound, zinc-bound, and cadmium-bound forms of the aqMutL CTD.

was not altered by the presence of cadmium ions, suggesting that the catalytic step of the endonucleolytic reaction was specifically disrupted by the ions (Fig S4A and B).

We previously found that cadmium ions bound to the same sites as zinc ions in the aqMutL CTD (27, 37). The gross structure of the cadmium-bound form of the aqMutL CTD was almost identical to that of the zinc-bound form. However, reanalysis of the detailed structures revealed that there were some differences between the two forms: High-resolution structures enabled us to find that the electron density for the guanidino group of Arg406 was missing in the zinc-bound form, whereas that was clearly seen in the cadmium-bound form (Fig 3C and D). The side chain of Arg406 was fixed by the M1 cadmium ion via a water molecule, W1. The distance between the M1 cadmium ion or zinc ion and the W1 water molecule was 2.53 or 2.24 Å, respectively (Fig 3E and F). In the cadmium-bound form, the side chain of Arg406 also interacted with the main-chain carbonyl group of Gly422 from the other subunit of the dimer (Fig 3E), causing a movement of the $\alpha4$ helix (Figs 3G and S5). The movement of the $\alpha4$ helix accompanied a change in the subunit arrangement of the aqMutL CTD dimer (Fig 3H and I). As the result, in the cadmium-bound form, Gly422 (carbonyl oxygen atom) was ~1.0 Å closer to Arg406 (C$\delta$ atom) than in the zinc-bound form (Fig S6A). Despite these structural changes, no significant difference was found in the structure around the metal-binding site except for the side chain of Arg406 (Fig S7).

To confirm that the observed differences depend on the metal species, cadmium ions in the crystal were replaced with zinc ions. The crystal of the cadmium-bound form was washed with and soaked in the crystallization buffer containing 50 mM $Zn^{2+}$ for 2 d. The same process was repeated three times. Replacement of cadmium ions with zinc ions was confirmed by the zinc K-edge X-ray absorption fine structure analysis (Fig S1C). The zinc-restored structure of the aqMutL CTD was solved by Zn-SAD phasing (Table S1). The result demonstrated that the electron density for the guanidino group of Arg406 was missing in the zinc-restored form (Fig 4A, left panel) and that the distance between the M1 zinc ion and the W1 water molecule was 2.24 Å (Fig 4A, right panel). The position of the $\alpha4$ helix in the zinc-restored form was identical to that in the zinc-bound (zinc-cocrystallized) form (Fig 4B). Furthermore, the structure of Arg406 and the quaternary structure of the zinc-restored form were also identical to those of the zinc-bound form (Figs S6B and S8, respectively). This is in good agreement with the previous report that inhibition of human MutL$\alpha$ by cadmium ion could be rescued by addition of zinc ion (40).

The electron density of the guanidino group of Arg406 was also missing in the manganese-bound form of the aqMutL CTD (Fig 4C, left panel and Fig S6E). The distance between the M1 manganese ion and the W1 water molecule was 2.25 Å (Fig 4C, right panel). Likewise, the position of the $\alpha4$ helix, the structure of Arg406, and the quaternary structure of the manganese-bound form were the same as those in the zinc-bound form (Figs 4D, S6C, and S8, respectively). These are consistent with the fact that manganese is catalytically competent in the endonuclease activity of MutL. However, because no structural differences were observed between the manganese-bound and zinc-bound forms, it could not be explained why manganese activates the activity more strongly than zinc.

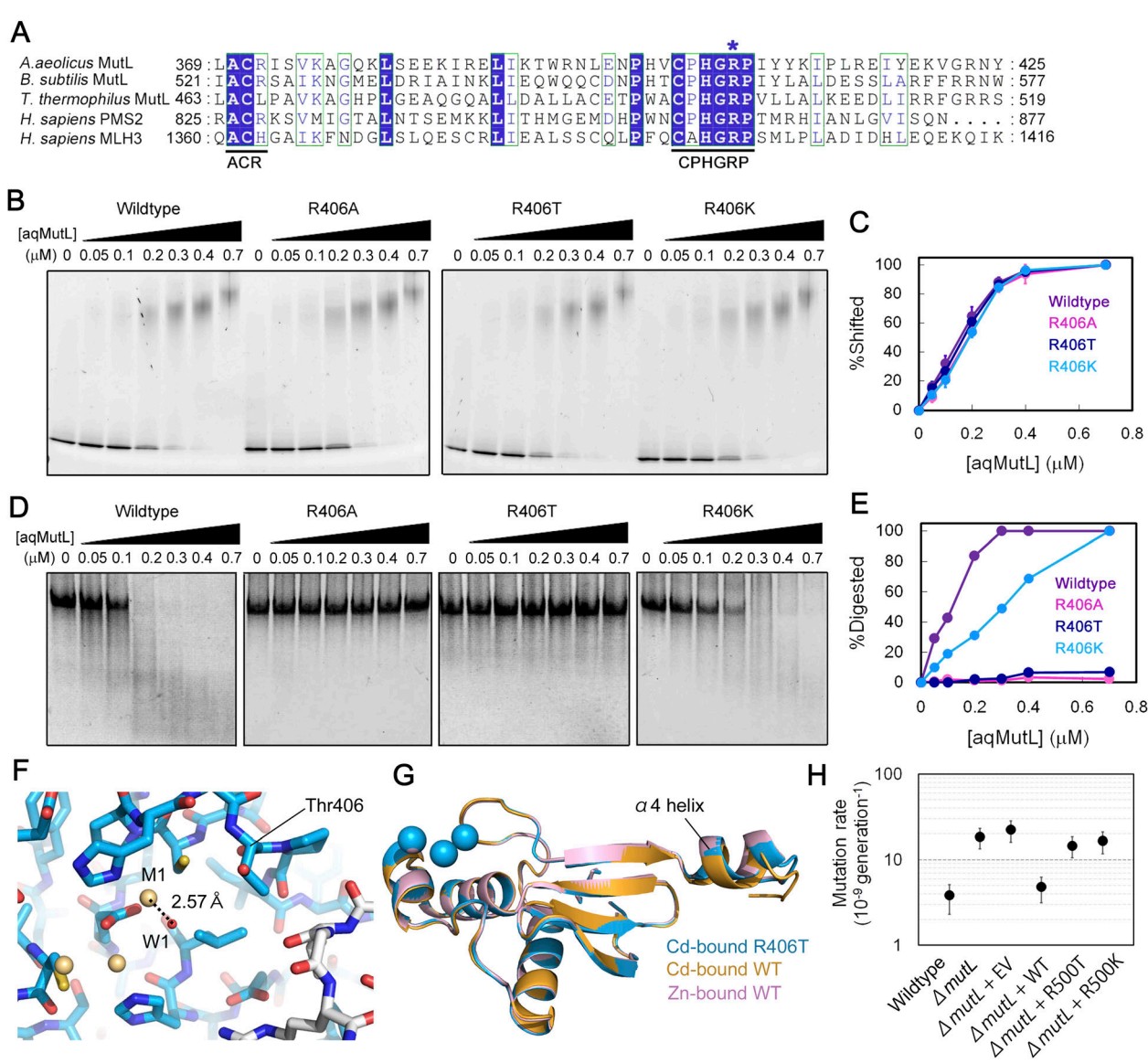

**Figure 5. Arg406 is a catalytic residue for the endonuclease activity of *A. aeolicus* MutL (aqMutL).**
**(A)** Amino acid sequence alignment of the C-terminal region of MutL endonucleases. The ACR and CPHGRP motifs are underlined. The conserved arginine residue corresponding to Arg406 of aqMutL is indicated by an asterisk. **(B)** DNA-binding abilities of the WT and mutant forms of the full-length aqMutL were examined by electrophoretic mobility shift assay. **(B, C)** Percentages of the shifted DNA signals in panel (B) were plotted against protein concentrations. Average values from three independent experiments are shown with standard deviations. **(D)** The endonuclease activity of the WT and mutant forms of aqMutL. A 60-bp double-stranded DNA was incubated with the proteins at 55°C for 60 min, and the incised products were separated by polyacrylamide gel electrophoresis under denaturing conditions. **(D, E)** The ratios of digested DNA signals to all DNA signals in panel (D) were plotted against protein concentrations. **(F)** The catalytic site of the cadmium-bound form of the R406T aqMutL C-terminal endonuclease domain (CTD). **(G)** Superimposition of the overall structure of the cadmium-bound form of the R406T aqMutL CTD with those of the cadmium-bound and zinc-bound forms of the WT aqMutL CTD. **(H)** Phenotype complementation experiments using *T. thermophilus*. Mutation rates of *T. thermophilus* strains were estimated by the fluctuation assay based on the frequency of streptomycin-resistant cells. Experiments were performed with 17 independent cultures for each strain. Bars indicate 95% confidence intervals. EV, empty vector.
Source data are available for this figure.

## Arg406 is the catalytic residue for the endonuclease activity of aqMutL

Arg406 is a component of the CPHGRP motif (Figs 1B and 5A) and completely conserved in all MutL endonucleases including human PMS2 and MLH3, the nuclease subunits of MutLα and MutLγ, respectively. Arg406 of aqMutL corresponds to Arg847 of human PMS2.

The R847T mutation has been found in Lynch syndrome-suspicious patients and registered as a variant of uncertain significance at the ClinVar database (48). In order to test the importance of Arg406 in the endonuclease activity, the R406A, R406T, and R406K mutant forms of the aqMutL were prepared. The circular dichroism spectra of the mutant forms were identical to that of the WT aqMutL (Fig S9), indicating that the mutations did not affect the integrity of the

protein structure. The DNA-binding ability of aqMutL was not influenced by these mutations (Fig 5B and C), whereas the endonuclease activity was severely impaired by the R406A and R406T mutations (Fig 5D and E). The R406K mutant form retained significant, albeit weaker, endonuclease activity. These results suggest that the Arg residue in the CPHGRP motif is a catalytic residue for the endonuclease activity.

The R406T mutant form of the aqMutL CTD was crystallized in the presence of cadmium ion, and the structure was solved with the Cd-SAD phasing (Table S1). As was expected from the CD measurements, the overall structure was almost identical to that of the WT aqMutL CTD. The distance between the M1 cadmium ion and the W1 water molecule was 2.57 Å (Fig 5F). However, the position of the $\alpha$4 helix, the local structure around Thr406, and the quaternary structure were the same as those of the zinc-bound form of the WT aqMutL CTD (Figs 5G, S6D, and S8, respectively). These results support the notion that the M1 cadmium ion fixes the side chain of Arg406 of aqMutL, causing the shift in the position of the $\alpha$4 helix and the change in the quaternary structure.

The importance of Arg406 for the repair activity in the cell was examined by phenotype complementation experiments using a eubacterium, *T. thermophilus*, one of the model organisms that have been used for studies on DNA repair including MMR (49). Disfunction of MMR causes a significant increase in mutation rate that can be determined by measuring the frequency of the streptomycin-resistant bacterial cells. The CPHGRP and other metal-binding motifs are also conserved in *T. thermophilus* MutL, with Arg500 corresponding to Arg406 of aqMutL. As shown in Fig 5H, introduction of the gene for the R500T *T. thermophilus* MutL did not complement the hypermutator phenotype of the *mutL*-lacking strain, indicating that the Arg residue in the CPHGRP motif is essential for in vivo MMR. This is consistent with a previous report that the replacement of the corresponding Arg residue with Glu abolished the in vitro MMR activity in the human cell extract (34). Thus, the result of the in vivo complementation experiment reinforces the interpretation that this Arg residue plays a catalytic role in the endonuclease activity of MutL. The gene encoding the R500K *T. thermophilus* MutL did not complement the hypermutator phenotype (Fig 5H), although the corresponding mutant form, the R406K, of aqMutL retained the endonuclease activity. Even a slight decrease in the endonuclease activity of MutL might be sufficient to abolish the in vivo MMR activity of *T. thermophilus*, which is consistent with the fact that an Arg residue at this position is conserved in MutLs from many biological species.

## Catalytic mechanism of the MutL endonuclease activity

The present crystallographic and other experiments suggest that the endonuclease reaction of MutL is catalyzed by the two-metal-ion mechanism, where metal ions at the M1 and M2 positions play a central role to cleave the phosphodiester bond of DNA backbone. Although the Asp residue of the DQHAX$_2$EX$_4$E motif has been known to be critical for the endonuclease activity, our crystallographic results revealed that the residue was not involved in the coordination of the catalytic metal ions. As previously suggested by Guarné and Charbonnier (38), Asp351 might be indirectly involved in

the reaction by stabilizing the DQHAX$_2$EX$_4$E motif-containing helix through the N-capping mechanism.

Results of both in vitro and in vivo experiments indicated that Arg406 is a catalytic residue in the endonuclease activity of aqMutL. Because the R406K mutant form of aqMutL retained significant endonuclease activity, it is thought that a positively charged amino acid residue at the position is important for the activity. Arg406 may have a role as a Lewis acid that, along with the catalytic zinc ions, stabilizes the negatively charged transition state generated by the attack of a nucleophilic water molecule. Further structural analyses on the MutL-DNA complex would help address the details of the role of the Arg residue.

## A mechanistic insight into the inhibitory effect of cadmium

In the two-metal ion mechanism, water molecules that are coordinated by the metal ions participate in the catalysis. The reactivity of the coordinated water molecules depends on the valence and radius of the coordinating metal ions. For metal ions of the same valence, the smaller the ionic radius, the smaller the p$K_a$ value of the coordinated water molecule, hence the stronger the nucleophile. The radius of hexacoordinated $Cd^{2+}$ (0.95 Å) is larger than that of hexacoordinated $Zn^{2+}$ (0.74 Å), which might explain the inhibitory effect of cadmium on the endonuclease activity of MutL. However, this idea seems to be inconsistent with the fact that $Mn^{2+}$ acts as a much stronger activator than $Zn^{2+}$ while having a larger ionic radius (0.83 Å) than $Zn^{2+}$.

Alternatively, the inhibitory effect of cadmium might be explained by the cadmium-dependent structural changes in the catalytic site. In the present study, it was confirmed that a cadmium ion fixed the guanidino group of a catalytic Arg residue of aqMutL. Furthermore, the fixed side chain of the residue interacted with a Gly residue, pulling the $\alpha$4 helix of the other subunit. The positional change of the $\alpha$4 helix only slightly affected the structure around the metal-binding site because of a concomitant change in the quaternary structure (Fig S6A). The restricted flexibility of the side chain of the catalytic Arg residue could be responsible for the inhibitory effect of cadmium.

Interestingly, we found that the R406K mutant form of aqMutL was resistant to the inhibitory effect of cadmium (Fig S10). ICP-AES measurements confirmed that zinc ions were completely replaced with cadmium ions in the R406K mutant form, as in the WT aqMutL (Table S2). The activity of the cadmium-bound form of the R406K aqMutL decreased to, but still retained, 60% of that of its zinc-bound form, indicating that cadmium is not optimal but inherently functional as a catalytic metal ion of aqMutL. This finding further supports the structural mechanism for the inhibitory effect of cadmium. Because the side chain of the Lys residue is shorter than that of the Arg residue and lacks the guanidino group, it is speculated that the binding of cadmium ions neither fixes the Lys residue in the same way as it does the Arg residue nor alters the position of the $\alpha$4 helix because the Lys residue cannot bridge the cadmium ion and Gly422. If MutL had evolved with a Lys residue at this position as the catalytic residue, MutL might not have been inhibited by cadmium. Although the structure of the cadmium-bound R406K aqMutL CTD should be helpful to confirm these

speculations, we could not obtain the crystal of this mutant form of CTD under the conditions examined.

# Materials and Methods

## Plasmid constructions and preparation of proteins

The expression plasmids for the R406A, R406T, and R406K forms of full-length aqMutL were constructed by introducing the mutations into the pET-11a/aqMutL plasmid using PrimeSTAR mutagenesis (Takara). The primer sets used were 5′-CACGGAGCACCCATATACTACAAAAT-3′ and 5′-TATGGGTGCTCCGTGGGGGCAAACGT-3′, 5′-CACGGAACACCCATATACTACAAAAT-3′ and 5′-TATGGGTGTTCCGTGGGGGCAAACGT-3′, and 5′-CACGGAAAACCCATATACTACAAAAT-3′ and 5′-TATGGGTTTTCCGTGGGGGCAAACGT-3′, respectively. The expression plasmids for the D366A, E384A, E416A, and R406T forms of the aqMutL CTD were constructed by the same procedure using the pET-11a/aqMutL CTD plasmid as the template. The primer sets used were 5′-CTGAAGGCCGAAAACTTAGCCTGCAG-3′ and 5′-GTTTTCGGCCTTCAGTTTTTCGTAGT-3′, 5′-TCGGAAGCGAAAATCAGAGAACTCAT-3′ and 5′-GATTTTCGCTTCCGAGAGCTTTTGTC-3′, 5′-CTGAGGGCAATATACGAAAAAGTAGG-3′ and 5′-GTATATTGCCCTCAGGGGTATTTTGT-3′, and 5′-CACGGAACACCCATATACTACAAAAT-3′ and 5′-TATGGGTGTTCCGTGGGGGCAAACGT-3′, respectively. DNA sequencing revealed that constructions were error free.

The WT and mutant forms of the full-length aqMutL and the aqMutL CTD were overexpressed in *E. coli* and purified by the previously described procedures (27). The protein solutions were concentrated and stored at 4°C.

## Crystallography of the manganese-bound form of the aqMutL CTD

The purified aqMutL CTD (20 $\mu$M) was incubated in the buffer containing 50 mM HEPES-KOH (pH 7.5), 100 mM NaCl, 1 mM DTT, 10% glycerol, and 2 mM $MnCl_2$ at 70°C for 24 h. After dialysis against the buffer containing 50 mM HEPES-KOH (pH 7.5), 10% glycerol, and 2 mM $MnCl_2$ at 25°C for 24 h, the protein was concentrated to 28.5 mg/ml. A 1 $\mu$l aliquot of the protein solution was mixed with an equal volume of the crystallization buffer containing 200 mM sodium acetate (pH 4.6), 50 mM $MnCl_2$, and 30% (vol/vol) polyethylene glycol 400. The drop was equilibrated against 50 $\mu$l of the same buffer at 20°C for 2 d. The crystal of the aqMutL CTD was cryocooled at −173°C. The diffraction data were collected at the wavelength 1.8920 Å for the manganese single-wavelength anomalous dispersion phasing and 1.0000 Å for the refinement of the model structure at SPring-8 BL38B1 (Hyogo). The diffraction data were processed by the HKL2000 program package (50). Identification of the locations of manganese atoms, initial phasing, phase improvements by density modification, and model building were performed using diffraction data of 1.8920 Å wavelength by PHENIX program (51). The model was refined using diffraction data of 1.0000 Å wavelength by COOT (52) and PHENIX programs. The statistics for data collection and refinement are shown in Table S1.

## Crystallography of the zinc-restored form of the aqMutL CTD

The crystal of the cadmium-bound form of the aqMutL CTD was obtained by the previously described Method (27). The cadmium-bound crystal was washed with the buffer containing 200 mM sodium acetate (pH 4.6) and 30% (vol/vol) polyethylene glycol 200 and soaked in the buffer containing 200 mM sodium acetate (pH 4.6), 50 mM $ZnCl_2$, and 30% (vol/vol) polyethylene glycol 400 for 2 d at 20°C. The same process was repeated three times. The crystal was cryocooled at −173°C, and the diffraction data were collected at the wavelength 1.2820 Å for the Zn-SAD phasing and 1.0000 Å for the refinement of the model structure at SPring-8 BL38B1. The data processing and refinement procedures were the same as those for the manganese-bound crystal of the aqMutL CTD.

## Crystallography of the cadmium-bound form of the R406T aqMutL CTD

The cadmium-bound form of the R406T aqMutL CTD was created by the same procedure as that for the WT aqMutL CTD (27). The crystal was cryocooled at −173°C, and the diffraction data were collected at the wavelength 1.5000 Å for the Cd-SAD phasing and 1.0000 Å for the refinement of the model structure at SPring-8 BL38B1. The data processing and refinement procedures were the same as those for the manganese-bound crystal of the aqMutL CTD.

## Inhibition of the endonuclease activity of aqMutL by cadmium

The WT or R406K mutant form of aqMutL (50 $\mu$M) was incubated in the buffer containing 50 mM HEPES-KOH (pH 7.5), 100 mM NaCl, 1 mM DTT, 10% glycerol, and 0.5 mM $ZnCl_2$ or $CdCl_2$ at 60°C for 0, 3, 6, 12, or 24 h. The supercoiled plasmid DNA (pT7Blue) (50 ng/$\mu$l) was incubated with 0–5 $\mu$M aqMutL of the WT or R406K mutant form in 50 mM HEPES-KOH (pH 7.5), 100 mM NaCl, 1 mM DTT, and 10% glycerol at 70°C for 30 or 15 min. Protein concentrations used in each experiment were indicated in the panel or the legends for figures. The reactions were stopped by adding 5 × loading buffer (10 mM EDTA, 1% SDS, 50% glycerol, and 0.05% bromophenol blue). Reaction solutions were loaded onto a 1.0% agarose gel containing TBE buffer and electrophoresed with the same buffer. DNA fragments were stained with SYBR Gold and detected under ultraviolet light at 365 nm. The amounts of the signals for the intact supercoiled and the incised DNAs were measured by using CS Analyzer 3.0 (ATTO).

Exchange of zinc for cadmium in the experiment was confirmed by ICP-AES (Table S2). Proteins (50 $\mu$M) were incubated with $ZnCl_2$ or $CdCl_2$ under the same condition as above for 24 h and dialyzed against the buffer containing 50 mM Tris–HCl (pH 7.5), 100 mM NaCl, 1 mM DTT, and 10% glycerol. After filtration, protein concentrations were determined on the basis of the values of absorbance at 280 nm, which indicated that precipitation of the protein during the manipulation was negligible. The concentrations of zinc and cadmium in the protein solutions were measured by an ICP-emission spectrometer PS3520UVDD II (Hitachi High-Tec).

### Assay of the endonuclease activity of the D366A, E384A, and E417A mutant forms of the aqMutL CTD using the plasmid DNA

In order to examine the manganese-dependent endonuclease activity of the aqMutL CTD, 50 ng/$\mu$l supercoiled plasmid DNA (pT7Blue) was incubated with 0–3 $\mu$M of the WT or the mutant forms of the aqMutL CTD in the buffer containing 50 mM HEPES-KOH (pH 7.5), 100 mM NaCl, 1 mM DTT, 10% glycerol, and 0.5 mM MnCl$_2$ at 60°C for 10 min. The amounts of the signals for the intact supercoiled and the incised DNAs were measured by the same procedure as above.

### Assay of the endonuclease activity of the R406A, R406T, and R406K mutant forms of aqMutL using the linear double-stranded DNA

The 5'-fluorescein-labeled 60-mer single-stranded DNA (5'-AGGAACCTCGAGGGATCCGTCCTAGCAAGCCGCTAGAGGAACATATCCTTAA GAGTTCCA-3') (eurofins Genomics) was hybridized with the non-labeled complementary strand (5'-TGGAACTCTTAAGGATATGTTCC TCTAGCGGCTTGCTAGGACGGATCCCTCGAGGTTCCT-3') to obtain the labeled 60-bp double-stranded DNA. The labeled DNA was incubated with 0–700 nM of the WT or the mutant forms of aqMutL in the buffer containing 50 mM HEPES-KOH (pH 7.5), 100 mM NaCl, 0.5 mM MnCl$_2$, 1.25 mM MgCl$_2$, 1 mM DTT, and 10% glycerol at 55°C for 60 min. The reaction was stopped by the addition of an equal volume of the sample buffer containing 5 mM EDTA, 80% deionized formamide, 10 mM NaOH, 0.1% bromophenol blue, and 0.1% xylene cyanol. The reaction mixtures were loaded onto a 15% acrylamide gel containing 8 M urea and TBE buffer and electrophoresed with the same buffer. DNAs were visualized by using LuminoGraph I (ATTO). The intensities of the signals for the intact and incised DNAs were determined by CS Analyzer 3.0.

### Electrophoretic mobility shift assay

The fluorescently labeled double-stranded DNA was generated by hybridizing the 5' Cy5-labeled 60-mer single-stranded DNA (5'-AGGAACCTCGAGGGATCCGTCCTAGCAAGCCGCTGCTACCGGAAGCTTCTC-GAGGTTCCT-3') (BEX Co.) with the non-labeled 60-mer single-stranded DNA (5'-AGGAACCTCGAGAAGCTTCCGGTAGCAGCGGCTTGCTA GGACGGATCCCTCGAGGTTCCT-3') (BEX Co.). The DNA (50 nM) was incubated with 0–0.7 $\mu$M aqMutL in the buffer containing 50 mM HEPES-KOH (pH 7.5), 100 mM NaCl, 2.5 mM MgCl$_2$, 0.1 mM DTT, and 10% glycerol at room temperature for 15 min. The mixtures were loaded onto a 10–20% gradient polyacrylamide gel (ATTO) and electrophorezed in EzRun TG buffer (ATTO). DNAs were visualized by using LuminoGraph I. The intensities of the signals were determined by CS Analyzer 3.0. The %Shifted values were calculated as percentage of shifted signals to all signals in each lane, i.e., %Shifted = 100 × $S_{shifted}/(S_{free} + S_{shifted})$, where $S_{shifted}$ and $S_{free}$ are the amounts of shifted and free DNA signals in each lane.

### Circular dichroism spectrometry

Circular dichroism measurements were carried out with a Jasco spectropolarimeter, model J-720W (Jasco). Measurements were performed in a solution containing 30 mM Tris–HCl (pH 8.0) and 10 $\mu$M protein using a 0.1 cm cell at 25°C. The residue molar ellipticity [$\theta$] was defined as $100\theta_{obs}/(lc)$, where $\theta_{obs}$ is the observed ellipticity, $l$ is the length of the light path in centimeters, and $c$ is the residue molar concentration of the protein.

### MS of the endonucleolytic products

The 16-mer single-stranded DNA (5'-CGGTATCTCGACTATG-3') (eurofins Genomics) was hybridized with the complementary strand (5'-CATAGTCGAGATACCG -3') to obtain the 16-bp double-stranded DNA. The double-stranded DNA (35 $\mu$M) was digested with or without 1.9 $\mu$M aqMutL for 6 h at 37°C in 25 mM Tris–HCl (pH 8.0) and 0.5 mM MnCl$_2$.

The reactions were stopped by heat treatment at 98°C for 20 min. The digested fragments in the reaction mixtures were trapped by the SuperQ resin. After washing with water, the fragments were eluted with 0.75 M ammonium acetate. Eluted DNAs were dried up in tubes and redissolved in water just before the MS experiment. The DNA fragments were mixed with 50% acetonitrile containing 10 mg/ml 3-hydroxypicolinic acid and loaded on a sample plate. MS spectra were measured using a RapifleX, MALDI TOF/TOF instrument (Bruker Daltonics Inc.). The m/z values were externally calibrated by peptide II and protein I standards (Bruker Daltonics Inc.).

### Complementation experiment with fluctuation assay

*T. thermophilus* HB8 strain was grown at 70°C in TR medium: 0.4% tryptone (Difco), 0.2% yeast extract (Oriental Yeast), and 0.1% NaCl (pH 7.5). The TR medium was supplemented with 0.4 mM CaCl$_2$ and 0.4 mM MgCl$_2$ (TT medium) when used for transformation experiments. The *mutL*-lacking strain of *T. thermophilus* HB8 was constructed as previously described (35). The WT, R500T, and R500K *T. thermophilus mutL* genes were ligated into the pMK18::HygΔKan plasmid using XbaI and HindIII restriction sites. The *mutL*-lacking strain of *T. thermophilus* HB8 was transformed with these plasmids. The mutation rate of *T. thermophilus* HB8 cells was estimated by the Luria-Delbruck fluctuation assay as described previously (53, 54). Single colonies of *T. thermophilus* HB8 on TR plates were inoculated into TR media and cultured at 70°C overnight. The overnight culture was diluted 500-fold with TR medium. The diluted culture (100 $\mu$l) was inoculated into 4 ml of TR medium and further incubated at 70°C overnight. The overnight culture was diluted 500-fold with TR medium. Aliquots of the diluted culture (1.5 ml) were dispensed into 20 test tubes and incubated at 70°C for 24 h. Three of the 20 cultures were diluted $10^5$-fold. The diluted culture (50 $\mu$l) was plated on the non-selective TR plate and incubated at 70°C for 16 h. The colony-forming units were counted to determine the final population size (Nt). The rest of the 20 cultures were plated on the selective TR plate containing 100 $\mu$g/ml streptomycin and incubated at 70°C for 30 h. The colony-forming units were counted, and the number of mutation events (m) were calculated by the program rSalvador (55). The mutation rate of each strain was estimated as m/Nt.

# Supplementary Information

# Acknowledgements

We thank beamline scientists at SPring-8 for help during data collection, Dr. Farid A Kadyrov (Southern Illinois University) for his advice on this work, and Dr. Yuki Fujii (Osaka City University) for help in the in vivo complementation experiment using *T. thermophilus* HB8. This work was supported by JSPS KAKENHI grant number JP19K07376 (to K Fukui). The crystallographic experiments were performed at BL38B1 in SPring-8 (JASRI proposal nos. 2016A2522 and 2016B2522).

## Author Contributions

K Fukui: conceptualization, data curation, funding acquisition, investigation, visualization, methodology, project administration, and writing—original draft, review, and editing.
T Yamamoto: investigation, methodology, and writing—review and editing.
T Murakawa: methodology and writing—review and editing.
S Baba: methodology and writing—review and editing.
T Kumasaka: methodology and writing—review and editing.
T Yano: conceptualization, supervision, project administration, and writing—original draft, review, and editing.

## Conflict of Interest Statement

The authors declare that they have no conflict of interest.

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
