## [Reviewer comments · Life Science Alliance]

Life Science Alliance

Catalytic mechanism of the zinc-dependent MutL endonuclease reaction

Kenji Fukui, Tatsuya Yamamoto, Takeshi Murakawa, Seiki Baba, Takashi Kumasaka, and Takato Yano

DOI: <https://doi.org/10.26508/lsa.202302001>

Corresponding author(s): Kenji Fukui, Osaka Medical and Pharmaceutical University and Takato Yano, Osaka Medical and Pharmaceutical University

Review Timeline:

Submission Date:	2023-02-19
Editorial Decision:	2023-03-17
Revision Received:	2023-06-28
Editorial Decision:	2023-07-10
Revision Received:	2023-07-11
Accepted:	2023-07-12

Scientific Editor: Novella Guidi

Transaction Report:

March 17, 2023

Re: Life Science Alliance manuscript #LSA-2023-02001-T

Dr. Kenji Fukui
Osaka Medical and Pharmaceutical University
Department of Biochemistry
2-7
Daigaku-machi
Takatsuki, Osaka 569-8686
Japan

Dear Dr. Fukui,

Thank you for submitting your manuscript entitled "Catalytic mechanism of the zinc-dependent MutL endonuclease reaction" to Life Science Alliance. The manuscript was assessed by expert reviewers, whose comments are appended to this letter. We invite you to submit a revised manuscript addressing the Reviewer comments.

Thank you for this interesting contribution to Life Science Alliance. We are looking forward to receiving your revised manuscript.

Sincerely,

B. MANUSCRIPT ORGANIZATION AND FORMATTING:

Reviewer #1 (Comments to the Authors (Required)):

Several labs, including that of the authors, have shown that MutL proteins of eukaryotes and many bacteria function as latent endonucleases that copurify and/or bind several zinc ions within a highly conserved endonuclease active site. Although biological activation of the MutL endonuclease depends on mismatch recognition by the cognate MutS protein and physical interaction with the corresponding replication clamp, endonuclease function of these proteins can be constitutively activated in the absence of other proteins by Mn²⁺, which has proven to be a useful tool for study of these enzymes. Furthermore, the mutagenic activity of the environmental carcinogen Cd²⁺, a known inhibitor of many zinc enzymes, has been attributed to its inhibition of mismatch repair, and in particular the MutL endonuclease.

Using structural and biochemical approaches, the authors provide mechanistic explanations for these metal effects. Based on crystal structures of *Aquifex aeolicus* MutL (aqMutL), they show that Mn²⁺ and Cd²⁺ substitute for active site Zn²⁺ ions, binding at the same sites. They also show that binding of Mn²⁺ activates endonuclease activity whereas binding of Cd²⁺ strongly inhibits endonuclease action. Based on high resolution crystal structures, they demonstrate that this metal functional difference is associated with positional movement of the active site α 4 helix in the Cd²⁺ complex, which does not occur in Zn²⁺ or Mn²⁺ complexes. In the Cd²⁺ complex, the highly conserved Arg406 guanido group interacts with a Cd²⁺-bound water molecule. This permits the Arg residue to interact with carbonyl group of Gly422 of the other subunit of the enzyme resulting in 3 Å movement of the active site α 4 helix. The authors also show genetically that Arg406 is required for functional mismatch repair in the bacterial cell, confirming the importance of this conserved residue in endonuclease function.

This nice paper clarifies the mechanism of mismatch repair inhibition by the carcinogen cadmium. The experimental evidence for the authors' conclusions appears sound, and I recommend acceptance with only minor modification to address one citation error:

Pg2 - last paragraph states "The mutagenic effect of cadmium has been linked to specific inhibition of the MMR activity⁶" Reference 6 is not the correct reference for this statement. This was shown by Gordenin and colleagues (Nat Genet 34:326-329).

Reviewer #2 (Comments to the Authors (Required)):

Overall, I am very enthusiastic about this manuscript. Here the authors have explored the metal requirements for aqMutL cleavage using biochemistry and X-ray crystallography and have extensively mutagenized a highly conserved arginine at the active site. Their analyses have firmly established the nature of the MutL cleavage products and provide a very interesting set of data that addresses how these enzymes are inhibited by cadmium. In many of their experiments, they've done an excellent job of demonstrating that the structures that they have determined are bound to the correct metals through XAS and ICEP. I have a number of comments that I believe will strengthen the manuscript.

Major comments.

1. The mass spectrometry analysis of the MutL hydrolysis products that demonstrate generation of 5'-phosphates and 3'-hydroxyls is really an important experiment. Unfortunately, the description of the results is too abbreviated to allow readers to independently evaluate them. The authors should report the observed masses of fragments 1-10, report if these fragments have a 5'-phosphate or 3'-hydroxyl, and compare them to the expected mass. It would also be helpful if the authors directly stated that these masses were for the single-stranded hydrolysis fragments and not partially double-stranded products. Similarly, Figure S3 could be improved by (1) adding a second MS/MS spectrum for a fragment with a 5'-phosphate, (2) show a diagram of the original dsDNA and cleavage event giving rise to the fragment, (3) annotation of the masses of some of the labeled peaks, and (4) description of z1, w1, d2/w2, d3/w3, and w4 labels for those not well versed in mass spectrometry.

2. Given the similarities of the Zn- and Mn-bound structures, is it possible to speculate why manganese accelerates the cleavage reaction? If not, it would be useful for the authors to state that no obvious structural changes explain the kinetic changes.

3. I really think that the manuscript would be improved if the authors somehow merged Figs. 3 and 4 so that we could simultaneously inspect the density and active site structures of: (1) Zn-bound, (2) Cd-bound, (3) Mn-bound, (4) Zn-restored proteins.

4. From the displayed images, it is hard to envision the alpha4 helix conformational change shown in the ribbon diagram with the very small change of Gly422 (e.g. compare Fig 3E, Fig 3F). I wonder if adding a panel to show the change in the alpha4 helix orientation in some view that is intermediate between the active site figures and the ribbon diagrams. How much does this conformational change affect the orientations of the alpha4 side chains, for instance?

5. The authors have demonstrated that cadmium locks down the alpha4 helix through interactions involving R406. Intriguingly, this conformational change would be expected to dramatically affect the environment of D351 in the DQHAX2EX4E motif. If D351 plays a role in catalysis (rather than just alpha helix stabilization), how does Cd inhibition affect its environment? In extension to point #4 above, the authors need to better describe the impact of the alpha4 helix movement in the Cd-bound structure.

6. The fact that the R406K mutant is functional and only has a modest endonuclease defect is really surprising given the almost absolute conservation of the R406 residue across billions of years of evolution. The authors should test the in vivo effects of the equivalent R500K mutation in *T. thermophilus* to see whether the activity of this variant is sufficient to support mismatch repair in vivo as well.

7. On page 6, the authors speculate that the R406K mutant is able to avoid cadmium inhibition as it doesn't trap the alpha4 helix orientation like the wild-type protein. There is, however, a second possibility: the R406K mutant doesn't readily exchange Zn for Cd. The authors should confirm that their only partially inhibited R406K mutants are actually Cd-bound at levels equivalent to the wild-type protein.

Other comments

Page 3. "Peaks of electron density in the anomalous diffraction map at 1.8900 Å were clearly..." (1) Is the X-ray wavelength known to that level of precision? (2) This statement is a little cryptic. It would be better to say "Peaks of electron density in the anomalous diffraction map calculated from data collected at a wavelength of 1.8900 Å were clearly..." It might also be useful to comment that 1.89 Å was chosen to be above the Mn absorption edge so that the anomalous peaks correspond to Mn ions.

Page 3. "Thus, it was confirmed that manganese ions bind to the same sites of aqMutL as zinc ions, indicating that the M1 and M2 sites are the binding sites for the catalytic metal ion and that zinc ion is the physiological catalytic metal ion." I believe this statement goes too far. The data clearly show that Mn ions are binding at the Zn ion sites (in addition to the other sites that aren't relevant to catalysis). But the data indicating that M1 and M2 are catalytic sites is not due to the Mn binding, but extensive mutagenesis of these sites in aqMutL and homologs. And this data cannot speak to Zn being the physiological ion. These proteins were generated by overproduction in *E. coli*. Identifying the physiological ion bound in *A. aeolicus* MutL requires substantial additional experimentation that is beyond the scope of this manuscript.

Page 4 paragraph 2. "It was recently found that cadmium was a specific inhibitor of the endonuclease activity of human MutLalpha by competing with zinc ions, which explains the carcinogenic effect of cadmium in human<s>." I also believe that this statement goes too far. Modrich did effectively demonstrate cadmium-mediated inhibition of MutLalpha, but cadmium had pleiotropic effects on organisms. Thus, it's unlikely that the only carcinogenic effect of cadmium is MutLalpha inhibition, even if MutLalpha inhibition is a contributing factor.

Page 4 paragraph 3. The authors are interpreting the gradual inhibition of aqMutL by incubation with cadmium as due to a Zn/Cd exchange. There are other possibilities, such as Cd-mediated precipitation of aqMutL. Further, the interpretation of the ion replacement is purely due to loss of activity. Do the authors have any independent measures of Zn/Cd exchange? Is aqMutL stable in the presence of Cd for 12 hours at 60 degrees?

Page 4 paragraph 3. How were the "cadmium-bound" aqMutL molecules generated for the DNA binding experiment in Fig S4? Were they taken from the final time point in the Cd exchange used for the kinetic measurements? How was the level of cadmium/zinc in these proteins verified?

Page 4 paragraph 4. Is Gly422 the C-terminal residue?

Page 4 paragraph 4. "...causing the C-terminal alpha4 helix to move by 2.70 A compared with the zinc-bound form." How was this 2.70 A distance measured? This should be included with the measurement. Clearly not all of the atoms in the helix have moved this much, as it appears to have something of a hinge at the N-terminus of the helix.

Page 5 paragraph 1. The sentence "This is in good agreement with the previous report that inhibition of MutLalpha by cadmium..." needs a reference added.

Figure 5H. It might be useful to draw horizontal lines through the rates of the wildtype and DmutL strains for comparison purposes.

Figure 5H legend. It would be useful for the authors to comment that the mutation rate assay they are using is streptomycin resistance.

Reviewer #3 (Comments to the Authors (Required)):

This manuscript describes crystal structures of the C-terminal domain (CTD) of *A. aeolicus* MutL with manganese, zinc and/or cadmium ions bound to its "CPHGRP" motif. The structural data are complemented by mass spectrometry data and mutation data in *T. thermophilus* in vivo that lead to the interpretation that the arginine in the CPHGRP motif has a two-metal catalytic role in the endonuclease activity of MutL, in a manner that can largely explain the inhibitory effect of cadmium on that activity.

Critique: The data are extensive, and they lead to two important points, namely that the endonuclease activity of MutL proteins requires two-metals, and that cadmium reduces the endonuclease activity to reduce the strand discrimination function for DNA mismatch repair by interfering with the this two-metal mechanism. Thus, the manuscript should be of interest to scientists interested in the mismatch repair mechanism, the origins of diseases and those interested in the biology of metals, including those of environmental concern. The crystallography is nicely complemented with biochemical and genetic data that support this major conclusions. I believe that this study is appropriate for publication, after revising the text in a few places, for both spelling and clarity. Chief among these are two minor but significant points. First, it would be highly appropriate to cite the paper published in 2003 in *Nature Genetics* by Jin et al, which is paper that set this and several earlier studies in motion. Secondly, it would be useful for non-experts to explain that eukaryotic MutL proteins are heterodimers in which only one subunit of the two subunits contains the endonuclease activity, and to slightly revise the first sentence of the next paragraph.

Our response to the Reviewer #1.

This nice paper clarifies the mechanism of mismatch repair inhibition by the carcinogen cadmium. The experimental evidence for the authors' conclusions appears sound, and I recommend acceptance with only minor modification to address one citation error:

Pg2 - last paragraph states "The mutagenic effect of cadmium has been linked to specific inhibition of the MMR activity" Reference 6 is not the correct reference for this statement. This was shown by Gordenin and colleagues (Nat Genet 34:326-329).

Response. We thank reviewer #1 for pointing out a citation error, which we have corrected in the revised manuscript (p. 2 line 41).

Our response to the Reviewer #2.

Major comments.

1. The mass spectrometry analysis of the MutL hydrolysis products that demonstrate generation of 5'-phosphates and 3'-hydroxyls is really an important experiment. Unfortunately, the description of the results is too abbreviated to allow readers to independently evaluate them. The authors should report the observed masses of fragments 1-10, report if these fragments have a 5'-phosphate or 3'-hydroxyl, and compare them to the expected mass. It would also be helpful if the authors directly stated that these masses were for the single-stranded hydrolysis fragments and not partially double-stranded products. Similarly, Figure S3 could be improved by (1) adding a second MS/MS spectrum for a fragment with a 5'-phosphate, (2) show a diagram of the original dsDNA and cleavage event giving rise to the fragment, (3) annotation of the masses of some of the labeled peaks, and (4) description of z1, w1, d2/w2, d3/w3, and w4 labels for those not well versed in mass spectrometry.

Response. We have added a new table, Table 1, in which the observed and theoretical masses of the fragments are listed, and the chemical group at the cleaved site of each fragment is also indicated. In the course of this revision, we identified a new fragment that we had previously missed and added it to Fig. 2 and Table 1 as the fragment #11. Fig. S3 has also been improved according to the reviewer's suggestion. First, a schematic representation for the original dsDNA and cleavage event giving rise to the parental ion is added in the inset. Second, annotation of the masses of the detected fragments are included in panel C. Finally, description of *a* to *d* and *w* to *z* ion species is also explained in panel B and figure legend. The fragments having a 5'-phosphate at the cleaved sites could not be fragmented by our MS/MS analysis; therefore, no MS/MS spectrum for the 5'-phosphate-containing fragment is shown in the manuscript. However, the presence of the 5'-phosphate and 3'-OH groups at the cleaved sites were confirmed by the parental MS spectra. The following sentences have been added in the revised manuscript:

“The cleavage products were completely dissociated into single-stranded DNAs during the MS analysis. The

masses of the detected fragments were consistent with the theoretical masses with errors of less than 100 ppm (Table 1), assuming that aqMutL leaves 5'-phosphate and 3'-OH groups at the cleavage site. Although fragments with 5'-phosphate group could not be efficiently fragmented by our MS/MS analysis, those with 3'-OH group were precisely identified by the MS/MS analysis (Supplementary Fig. S3)." (p. 4 line 7–13)

2. *Given the similarities of the Zn- and Mn-bound structures, is it possible to speculate why manganese accelerates the cleavage reaction? If not, it would be useful for the authors to state that no obvious structural changes explain the kinetic changes.*

Response. We agree with the reviewer's suggestion, and the following sentences have been added in the revised manuscript:

"These are consistent with the fact that manganese is catalytically competent in the endonuclease activity of MutL. However, since no structural differences were observed between the manganese-bound and zinc-bound forms, it could not be explained why manganese activates the activity more strongly than zinc." (p. 5 line 24–28)

3. *I really think that the manuscript would be improved if the authors somehow merged Figs. 3 and 4 so that we could simultaneously inspect the density and active site structures of: (1) Zn-bound, (2) Cd-bound, (3) Mn-bound, (4) Zn-restored proteins.*

Response. We have added Supplementary Fig. S6, where the Zn-bound structure was superimposed onto the Cd-bound form (panel A), Zn-restored form (panel B), Mn-bound form (panel C) of the wildtype aqMutL CTD, and Zn-bound form of the R406T aqMutL CTD (panel D). The electron density maps of these structures have also been shown side by side in panel E.

4. *From the displayed images, it is hard to envision the alpha4 helix conformational change shown in the ribbon diagram with the very small change of Gly422 (e.g. compare Fig 3E, Fig 3F). I wonder if adding a panel to show the change in the alpha4 helix orientation in some view that is intermediate between the active site figures and the ribbon diagrams. How much does this conformational change affect the orientations of the alpha4 side chains, for instance?*

Response. The original version of Fig. 3G was misleading due to our carelessness. In the original version, each subunit of the Cd-bound form was separately superimposed onto the corresponding subunit of the Zn-bound form. When displayed in that way, one would think that there is a significant difference in the active-site between the Zn-bound and Cd-bound forms. In fact, there was also a change in the quaternary structure: In the revised version, the Cd-bound dimeric structure is correctly superimposed onto the Zn-bound dimeric structure, which shows the difference in the quaternary structure between the two forms. The change in the quaternary structure, which is illustrated in the Fig. 3I, attenuated the effect of the positional change of the $\alpha 4$ helix (2.7 Å) on the structure of the catalytic site. As the result, the Cd-dependent change in distance between Arg406 (C δ atom) and Gly422 (carbonyl oxygen atom) was approximately 1 Å (Supplementary Fig. S6A). These have been described in the main text as follows:

"..., causing a movement of the $\alpha 4$ helix (Fig. 3G and Supplementary Fig. S5). The movement of the $\alpha 4$ helix accompanied a change in the subunit arrangement of the aqMutL CTD dimer (Fig. 3H and 3I). As the result, in the cadmium-bound form, Gly422 (carbonyl oxygen atom) was approximately 1.0 Å closer to Arg406 (C δ atom) than in the zinc-bound form (Supplementary Fig. S6A). Despite these structural changes, no significant difference was found in the structure around the metal binding site except for the side chain of Arg406 (Supplementary Fig. S7)." (p. 4 line 46–p. 5 line 3)

"Furthermore, the structure of Arg406 and the quaternary structure of the zinc-restored form were also identical to those of the zinc-bound form (Supplementary Fig. S6B and Supplementary Fig. S8, respectively)." (p. 5 line 14–16)

"Likewise, the position of the $\alpha 4$ helix, the structure of Arg406, and the quaternary structure of the manganese-bound form were the same as those in the zinc-bound form (Fig. 4D, Supplementary Fig. S6C, and Supplementary Fig. S8, respectively)." (p. 5 line 22–28)

“However, the position of the α 4 helix, the local structure around Thr406, and the quaternary structure were the same as those of the zinc-bound form of the wildtype aqMutL CTD (Fig. 5G, Supplementary Fig. S6D, and Supplementary Fig. S8, respectively). These results support the notion that the M1 cadmium ion fixes the side chain of Arg406 of aqMutL, causing the shift in the position of the α 4 helix and the change in the quaternary structure.” (p. 5 line 49–p. 6 line 5)

5. *The authors have demonstrated that cadmium locks down the alpha4 helix through interactions involving R406. Intriguingly, this conformational change would be expected to dramatically affect the environment of D351 in the DQHAX2EX4E motif. If D351 plays a role in catalysis (rather than just alpha helix stabilization), how does Cd inhibition affect its environment? In extension to point #4 above, the authors need to better describe the impact of the alpha4 helix movement in the Cd-bound structure.*

Response. We thank the reviewer for pointing this out. As noted above, most of the effect of the large movement of the α 4 helix was mitigated by the change in the quaternary structure. As the result, no structural change occurred around Asp351. To illustrate this, Supplementary Fig. S7 has been added.

6. *The fact that the R406K mutant is functional and only has a modest endonuclease defect is really surprising given the almost absolute conservation of the R406 residue across billions of years of evolution. The authors should test the in vivo effects of the equivalent R500K mutation in T. thermophilus to see whether the activity of this variant is sufficient to support mismatch repair in vivo as well.*

Response. According to the reviewer’s suggestion, we tested the effect of the R500K mutation on the *in vivo* MMR activity in *T. thermophilus*. As the result, the mutation impaired the intracellular repair activity to the same extent as the R500T mutation (Fig. 5H of the revised version). This might indicate that even a slight decrease in the endonuclease activity of MutL has serious effects on intracellular repair activity. The following sentences have been added:

“The gene encoding the R500K *T. thermophilus* MutL did not complement the hyper mutator phenotype (Fig. 5H), although the corresponding mutant form, the R406K, of aqMutL retained the endonuclease activity. Even a slight decrease in the endonuclease activity of MutL might be sufficient to abolish the *in vivo* MMR activity of *T. thermophilus*, which is consistent with the fact that an Arg residue at this position is conserved in MutLs from many biological species.” (p. 6 line 18–23)

7. *On page 6, the authors speculate that the R406K mutant is able to avoid cadmium inhibition as it doesn't trap the alpha4 helix orientation like the wild-type protein. There is, however, a second possibility: the R406K mutant doesn't readily exchange Zn for Cd. The authors should confirm that their only partially inhibited R406K mutants are actually Cd-bound at levels equivalent to the wild-type protein.*

Response. We have done an additional experiment to confirm that the R406K mutant form of aqMutL exchanges Zn for Cd as readily as the wildtype aqMutL. The wildtype and R406K mutant forms of aqMutL were incubated with Zn or Cd for 24 h under the same condition as the experiments in Fig. 3A and Supplementary Fig. S6 (Fig. S9 in the revised manuscript). After dialyzing the protein solutions against the metal-free buffer, the metal contents of the protein solutions were measured by ICP-AES. As the result, zinc ions were completely replaced by cadmium ions in the R406K mutant form as in the wildtype form. These results are shown in the Supplementary Table S2, and the following sentences have been added in the main text:

“ICP-AES measurements confirmed that zinc ions were completely replaced with cadmium ions in the R406K mutant form as in the wildtype aqMutL (Supplementary Table S2).” (p. 7 line 12–14)

“Exchange of zinc for cadmium in the experiment was confirmed by ICP-AES (Supplementary Table S2). Proteins (50 μ M) were incubated with ZnCl₂ or CdCl₂ under the same condition as above for 24 h, and dialyzed against the buffer containing 50 mM Tris-HCl (pH 7.5), 100 mM NaCl, 1 mM DTT, and 10 % glycerol. After filtration, protein concentrations were determined on the basis of the values of absorbance at 280 nm, which indicated that precipitation of the protein during the manipulation was negligible. The concentrations of zinc and cadmium in the protein solutions were measured by ICP emission spectrometer PS3520UVDD II (Hitachi High-Tec).” (p. 9 line 29–36).

Other comments

Page 3. "Peaks of electron density in the anomalous diffraction map at 1.8900 Å were clearly..." (1) Is the X-ray wavelength known to that level of precision?

Response. Yes. The scientific team at the SPring-8 BL26B1 beam line recommends that the wavelength be indicated to four decimal places.

(2) This statement is a little cryptic. It would be better to say "Peaks of electron density in the anomalous diffraction map calculated from data collected at a wavelength of 1.8900 Å were clearly..." It might also be useful to comment that 1.89 Å was chosen to be above the Mn absorption edge so that the anomalous peaks correspond to Mn ions.

Response. According to the reviewer's suggestion, we have corrected the sentence as follows:

"Peaks of electron density in the anomalous diffraction map calculated from data collected at a wavelength of 1.8900 Å were clearly..." (p. 3 line 32–33),

and added the following sentence:

"The wavelength of 1.8900 Å was chosen to be above the absorption edge of manganese so that the observed peaks of anomalous diffraction correspond to manganese atoms." (p. 3 line 34–36)

Page 3. "Thus, it was confirmed that manganese ions bind to the same sites of aqMutL as zinc ions, indicating that the M1 and M2 sites are the binding sites for the catalytic metal ion and that zinc ion is the physiological catalytic metal ion." I believe this statement goes too far. The data clearly show that Mn ions are binding at the Zn ion sites (in addition to the other sites that aren't relevant to catalysis). But the data indicating that M1 and M2 are catalytic sites is not due to the Mn binding, but extensive mutagenesis of these sites in aqMutL and homologs. And this data cannot speak to Zn being the physiological ion. These proteins were generated by overproduction in *E. coli*. Identifying the physiological ion bound in *A. aeolicus* MutL requires substantial additional experimentation that is beyond the scope of this manuscript.

Response. We agree with the reviewer and have corrected the description to avoid overstatement as follows:

"Thus, it was confirmed that manganese ions bind to the same sites of aqMutL as zinc ions. Together with experimental results from previous studies identifying catalytically important residues, the M1 and M2 sites are likely to be the binding sites for the catalytic metals." (p. 3 line 47–49)

Page 4 paragraph 2. "It was recently found that cadmium was a specific inhibitor of the endonuclease activity of human MutL α by competing with zinc ions, which explains the carcinogenic effect of cadmium in human cells." I also believe that this statement goes too far. Modrich did effectively demonstrate cadmium-mediated inhibition of MutL α , but cadmium had pleiotropic effects on organisms. Thus, it's unlikely that the only carcinogenic effect of cadmium is MutL α inhibition, even if MutL α inhibition is a contributing factor.

Response. We agree with this point and have corrected the sentence:

"It was recently found that cadmium was a specific inhibitor of the endonuclease activity of human MutL α by competing with zinc ions, which would be one of the mechanisms of the carcinogenic effect of cadmium." (p. 4 line 24–26)

Page 4 paragraph 3. The authors are interpreting the gradual inhibition of aqMutL by incubation with cadmium as due to a Zn/Cd exchange. There are other possibilities, such as Cd-mediated precipitation of aqMutL. Further, the interpretation of the ion replacement is purely due to loss of activity. Do the authors have any independent measures of Zn/Cd exchange? Is aqMutL stable in the presence of Cd for 12 hours at 60 degrees?

Response. In the ICP-AES measurement, no precipitation of aqMutL was observed after 60 °C heat treatment for 24

h with 0.5 mM CdCl₂ (mentioned in the revised manuscript on p. 9, line 29–36), although we found that aqMutL precipitated with cadmium at concentrations above 2 mM.

Page 4 paragraph 3. How were the "cadmium-bound" aqMutL molecules generated for the DNA binding experiment in Fig S4? Were they taken from the final time point in the Cd exchange used for the kinetic measurements? How was the level of cadmium/zinc in these proteins verified?

Response. The Cd-bound aqMutL used for the DNA-binding experiment was taken from the final time point in the Cd-exchange experiment in Fig. 3A. In the legend for Supplementary Fig. S4 of the revised version, the following explanation has been added:

“After treatment with 0.5 mM ZnCl₂ or CdCl₂ at 60 °C for 24 h, aqMutL was used for the assay.”

Page 4 paragraph 4. Is Gly422 the C-terminal residue?

Response. Gly422 is not at the C-terminus. The term “C-terminal” was removed from the sentence (p.4 line 46).

Page 4 paragraph 4. "...causing the C-terminal alpha4 helix to move by 2.70 Å compared with the zinc-bound form." How was this 2.70 Å distance measured? This should be included with the measurement. Clearly not all of the atoms in the helix have moved this much, as it appears to have something of a hinge at the N-terminus of the helix.

Response. When the single subunit of the zinc-bound form was superimposed onto that of the cadmium-bound form, the distance between the C α atoms of Val421 in the two forms was 2.70 Å. This has been illustrated in Supplementary Fig. S5 of the revised version.

Page 5 paragraph 1. The sentence "This is in good agreement with the previous report that inhibition of MutLalpha by cadmium..." needs a reference added.

Response. The reference (Sherrer *et al.* (2018) *PNAS* **115**, 7314-7319) has been indicated (p. 5 line 18).

Figure 5H. It might be useful to draw horizontal lines through the rates of the wildtype and DmutL strains for comparison purposes.

A horizontal line has been drawn in the revised version of Figure 5H.

Figure 5H legend. It would be useful for the authors to comment that the mutation rate assay they are using is streptomycin resistance.

Response. The following sentence has been added to the Figure 5H legend:

“Mutation rates of *T. thermophilus* strains were estimated by the fluctuation assay based on the frequency of streptomycin-resistant cells.”

Our response to the Reviewer #3.

*I believe that this study is appropriate for publication, after revising the text in a few places, for both spelling and clarity. Chief among these are two minor but significant points. First, it would be highly appropriate to cite the paper published in 2003 in Nature Genetics by Jin *et al.*, which is paper that set this and several earlier studies in motion. Secondly, it would be useful for non-experts to explain that eukaryotic MutL proteins are heterodimers in which only one subunit of the two subunits contains the endonuclease activity, and to slightly revise the first sentence of the next paragraph.*

Response. The reference #44, Jin *et al.* (2003) *Nature Genetics* **34**, 326-329, has been cited in the revised manuscript as follows:

“The mutagenic effect of cadmium has been linked to specific inhibition of the MMR activity⁴⁴.” (p. 2 line 40–41)

In the introduction section of the revised version, the following explanations have been added:

“Prokaryotic MutL endonucleases are homodimers, whereas eukaryotic MutL endonucleases are heterodimers with only one of the two subunits having the endonuclease activity.” (p. 1 line 43–45)

“Prokaryotic MutL endonucleases and the endonuclease subunits of eukaryotic MutL homologs include two domains.” (p. 2 line 3–4).

The manuscript has been thoroughly reviewed to eliminate spelling errors.

July 10, 2023

RE: Life Science Alliance Manuscript #LSA-2023-02001-TR

Dr. Kenji Fukui
Osaka Medical and Pharmaceutical University
Department of Biochemistry
2-7
Daigaku-machi
Takatsuki, Osaka 569-8686
Japan

Dear Dr. Fukui,

Thank you for submitting your revised manuscript entitled "Catalytic mechanism of the zinc-dependent MutL endonuclease reaction". We would be happy to publish your paper in Life Science Alliance pending final revisions necessary to meet our formatting guidelines.

- please upload all figure files as individual ones, including the supplementary figure files; all figure legends should only appear in the main manuscript file
- please add ORCID ID for the secondary corresponding author--they should have received instructions on how to do so
- please add the Twitter handle of your host institute/organization as well as your own or/and one of the authors in our system
- please add an Author Contributions section to your main manuscript text
- please add callouts for Figures S3A-C and S4A-B to your main manuscript text

A. FINAL FILES:

B. MANUSCRIPT ORGANIZATION AND FORMATTING:

Sincerely,

Reviewer #2 (Comments to the Authors (Required)):

The authors have done an excellent job of revising the manuscript and have satisfied all of my important concerns. The current manuscript is appropriate for publication.

July 12, 2023

RE: Life Science Alliance Manuscript #LSA-2023-02001-TRR

Dr. Kenji Fukui
Osaka Medical and Pharmaceutical University
Department of Biochemistry
2-7
Daigaku-machi
Takatsuki, Osaka 569-8686
Japan

Dear Dr. Fukui,

Thank you for submitting your Research Article entitled "Catalytic mechanism of the zinc-dependent MutL endonuclease reaction". It is a pleasure to let you know that your manuscript is now accepted for publication in Life Science Alliance. Congratulations on this interesting work.

DISTRIBUTION OF MATERIALS:

Again, congratulations on a very nice paper. I hope you found the review process to be constructive and are pleased with how the manuscript was handled editorially. We look forward to future exciting submissions from your lab.

Sincerely,
